# TMT and PRM Based Quantitative Proteomics to Explore the Protective Role and Mechanism of Iristectorin B in Stroke

**DOI:** 10.3390/ijms242015195

**Published:** 2023-10-15

**Authors:** Meizhu Zheng, Mi Zhou, Tingting Lu, Yao Lu, Peng Qin, Chunming Liu

**Affiliations:** 1College of Life Sciences, Changchun Normal University, Changchun 130032, China; zhengmz605@mail.cncnc.edu.cn; 2Central Laboratory, Changchun Normal University, Changchun 130032, Chinalutingting2580@163.com (T.L.); ly824219@163.com (Y.L.);

**Keywords:** Iristectorin B, stroke, ferroptosis, quantitative proteomics, tandem mass tag, parallel reaction monitoring

## Abstract

Stroke is a serious disease caused by the rupture or blockage of the cerebrovascular system. Its pathogenesis is complex and involves multiple mechanisms. Iristectorin B is a natural isoflavone that has certain anti stroke effects. In this study, an in vitro stroke injury model of glyoxylate deprivation was established using PC12 cells, which was used to evaluate the anti-stroke activity of Iristectorin B in ejecta stem. The results showed that Iristectorin B, a natural isoflavone derived from Dried Shoot, significantly reduced the damage to PC12 cells caused by oxygen glucose deprivation/reoxygenation, decreased apoptosis, enhanced cell survival and reduced Ca^2+^, LDH and ROS levels. The results showed that Iristectorin B had a significant protective effect on Na_2_S_2_O_4_-injured PC12 cells, and the mechanism may be related to the protective effect of neurons in the brain. After protein extraction and various analyses were performed, a series of cutting-edge technologies were organically combined to study the quantitative proteome of each group. Differential proteins were then analyzed. According to the protein screening principle, ferroptosis-related proteins were most closely associated with stroke. The differential proteins associated with ferroptosis screened were SLC3A2, TFR1 and HMOX1, with HMOX1 being the most significantly elevated and reduced via dosing. Iristectorin B may act as a protective agent against stroke by regulating ferroptosis, and SLC3A2, TFR1 and HMOX1 may serve as potential diagnostic biomarkers for stroke, providing additional evidence to support the importance of ferroptosis in stroke.

## 1. Introduction

Stroke is a serious disease caused by the rupture or blockage of the cerebrovascular system and poses a major health risk due to its high disability and mortality rates [1]. The onset of stroke leads to neuronal damage and nerve cell death, and subsequent ischemia-reperfusion injury (I/R) exacerbates cellular damage, leading to devastating and irreversible brain damage [2]. The pathogenesis of stroke is complex and involves multiple mechanisms, of which programmed neuronal cell death plays an important role. Ferroptosis is also strongly associated with the development of stroke [3]. The occurrence of stroke leads to iron overload and disturbances in lipid metabolism, with elevated iron levels causing lipid peroxidation and, ultimately, ferroptosis. Ferroptosis is a unique form of programmed cell death, regulated and influenced by multiple factors [4]. Ferroptosis is characterized by iron overload, glutathione (GSH) depletion, glutathione peroxidase (GPX4) inactivation and imbalances in lipid and amino acid metabolism [5]. Ferroptosis can be caused by various factors, mainly related to elevated levels of reactive oxygen species (ROS) in cells [6]. As the reactive oxygen species produced during ferroptosis accumulate, the cell will eventually move towards oxidative death. As science has evolved, more attention has been paid to the potential role of ferroptosis in stroke pathology. Numerous studies have demonstrated [7,8,9,10] that ferroptosis is closely related to the pathophysiological mechanisms of stroke, and its inhibition also reduces the secondary brain damage caused by the onset of a stroke, suggesting that ferroptosis is a potential therapeutic target for the treatment of strokes [11,12]. However, the effects of ferroptosis on the diagnosis, prognosis or treatment of stroke are not fully understood.

Proteomics is essentially the large-scale study of protein expression levels, protein–protein interactions and other protein signatures under different conditions [13], contributing to an integrated understanding of disease mechanisms and cellular metabolism at the protein level [14]. Tandem Mass Tag (TMT) technology is a quantitative in vitro peptide tagging technique [15]. Multiple stable isotope labels are used to perform tandem mass spectrometry analysis by specifically labeling the amino groups of peptides while comparing the relative amounts of proteins in a number of different samples. 

In this study, Iristectorin B is a natural isoflavone, which has certain anti-stroke effects. A large number of proteins were quantified via quantitative protein mass spectrometry with TMTs [16], and differentially expressed proteins associated with the onset and progression of stroke disease were identified. Our study may provide a theoretical basis for the search for biomarkers relevant to the diagnosis or treatment of stroke. 

## 2. Results

### 2.1. Cell Experimental Results

#### 2.1.1. Morphological Changes in PC12 Cells

After treatment with Na_2_S_2_O_4_ for 2 h, cell damage was seen in the model group under an inverted microscope, with reduced synapses, a rounded morphology and a significantly lower number of cells compared to the control group. In the presence of Iristectorin B or edaravone, the cells regained their original morphology, the cell damage improved and the number of viable cells increased (Figure 1).

#### 2.1.2. Effect of the Treatment of the PC12 Cells with Increasing Concentrations of Iristectorin B

As shown in Figure 2A, the survival rate of PC12 cells in the model group was significantly lower than in the normal group; compared to the model group, the survival rate of cells in the Iristectorin B 12.5 µg/mL dose groups was significantly increased.

As the results in Figure 2 show, the Ca^2+^, ROS, LDH level in the model group was significantly higher than that in the control group; compared to the model group, the Ca^2+^, ROS and LDH levels in the Iristectorin B 25 µg/mL dose groups were significantly low.

As shown in the flow cytometry apoptosis graph in Figure 3, the apoptosis rate was 3.6% in the control group and 22.17% in the model group. After OGD/R, the percentage of apoptotic cells in PC12 was significantly higher than that in the control group. The apoptosis rate was 14.89% in the Iristectorin B 6.25 µg/mL-treated group, 9.5% in the 12.5 µg/mL-treated group, 7.77% in the 25 µg/mL-treated group and 6.35% in the edaravone-positive-treated group. The apoptosis rate was significantly reduced in the drug and positive groups compared to the above two groups.

### 2.2. General Overview of Protein Identification

The results of the quantitative protein principal component analysis for all groups are presented in the below graph (Figure 4A). The degree of aggregation between samples in the graph represents the magnitude of variability in the samples, with closer three-point distances better representing protein reproducibility, with the red, blue and green groups all being denser. The total spectrums are displayed in Figure 4B. In total, 316,836 secondary spectra from mass spectrometry were identified; for matched spectrums, 113,311 spectra matched to theoretical secondary spectra were identified. For the peptides identified, 35,313 peptide sequences were resolved via matching; for unique peptides, 353,313 peptides were resolved via matching. For unique peptides, 33,541 unique peptide sequences were resolved via matching; for identified proteins, 6442 proteins were resolved via unique peptides. For quantifiable proteins, 6355 proteins were resolved via specific peptides. 

The ratio of the mean relative quantitative values of each protein in the multiple replicate samples was used as the fold change (FC) of the difference. To determine the significance of the differences, the relative quantitative values of each protein in the comparison group samples were subjected to a t-test, and the corresponding P-value was calculated as an indicator of significance, with the default being *p* < 0.05. To ensure the test data conformed to the normal distribution required by the t-test, the relative quantitative protein values were log2 transformed prior to the test. With the above analysis of variance, when *p* < 0.05, a change in differential expression of more than 1.3 was used as the threshold of change for significant upregulation, and less than 1/1.3 was the threshold of change for significant downregulation. The differential protein statistics plot (Figure 4C) and the differential protein volcano plot (Figure 4D) show red-colored dots, which represent elevated proteins, and green-colored dots, which represent decreased proteins. The differential protein heat map is presented in Figure 5. 

### 2.3. GO and KEGG Analysis of Differentially Expressed Proteins

The statistics of differential protein analysis were performed separately for biological processes (BP), cellular components (CC) and molecular functions (MF) (Figure 6A,B). The biological process analysis of the model/control group was performed first and identified 155 cellular transforming proteins, 118 proteins related to bioregulatory processes, 104 proteins related to metabolic processes and 88 proteins related to stress responses via screening. For cellular fraction analysis, 172 cell-associated proteins, 164 organelle-associated proteins and 60 protein complexes were screened. In the molecular functional analysis, 121 binding-related proteins and 60 catalytic activity-related proteins were observed. The biological process analysis in the experimental/model group was then performed, which identified 20 cellular transformation proteins, 16 metabolic process-related proteins, 14 bioregulatory-related proteins and 11 multicellular biological process-related proteins via screening. For cell composition analysis, 25 cell-associated proteins and 25 organelle-associated proteins were screened. In the molecular functional analysis, 19 binding-related proteins and 7 catalytic activity-related proteins were identified. Both experimental groups showed greater significant enrichment of ribosomal and ribosomal subunit-related proteins The analysis of molecular function-related proteins showed significant changes in cell growth factor and energy activity-related proteins in the model group conditions; the enrichment of nucleic acid DNA-related proteins was more significant in the experimental group after dosing compared to the model group. Both experimental groups showed greater significant enrichment of ribosomal and ribosomal subunit-related proteins, and the results are shown in Table 1, Table 2 and Table 3.

Analysis of the KEGG pathway for all differentially expressed proteins revealed significant changes in the ferroptosis pathway. In the detailed map of this pathway (Figure 7), HMOX1 was found to be the most highly associated protein via this pathway. Among the differential proteins identified in this study, HMOX1, TFR1, NCOA4 and SLC3A2 are the most important proteins involved in this pathway.

### 2.4. Bioinformatics Analysis 

#### 2.4.1. Subcellular Structures

The analysis of the main distribution of differential proteins at the subcellular level (shown in Figure 6C,D) was carried out. In the model/control group, 32.46% of the differential proteins were located in the nucleus, 22.51% were located in the cytoplasm, 8.9% were located in the mitochondria and 13.61% were located in the extracellular components. In the experimental/model group, 38.46% of the differential proteins were located in the nucleus, 19.23% were located in the cytoplasm, 19.22% were located in the mitochondria and 7.69% were located in the extracellular components. A large number of proteins differed between the nucleus, cytoplasm and mitochondria in both groups, suggesting that stroke protection mainly involves mechanisms in the nucleus, cytoplasm and mitochondria. 

#### 2.4.2. COG/KOG and Cluster Enrichment Analysis

The COG/KOG functional classification was performed for the differentially expressed proteins listed in Table 4. The majority of the differential proteins were signaling proteins, followed by information storage and processing related proteins, and some proteins with unknown functions were present. The functional categories and pathways in which differentially expressed proteins were significantly enriched (*p* < 0.05) are shown in bubble plots (Figure 8). Cluster analysis was performed to explore the correlation between differentially expressed protein functions in different comparison groups (Figure 9).

### 2.5. PPI Analysis

The differential protein interaction network was visualized using Cytoscape 3.9.0 software (Figure 10). Among the proteins that differ significantly in terms of their regulation of stroke through ferroptosis are TFR1, HMOX1 and SLC3A2. The upregulation of TFR1 indirectly affects the upregulation of NCOA4 through FTH1, and there is a link between the downregulation of AKT1 and the upregulation of HMOX1, which indirectly upregulates GSS expression through the downregulation of CAT. The circles in the diagram indicate differential proteins; green and red represent downregulated proteins and upregulated proteins, respectively; larger shapes show a greater number of interactions nodes; darker colors show larger differential ploidy; and triangles indicate the target differential proteins, with the gene names displayed. 

### 2.6. PRM Validation of Differential Proteins

Based on the results of the bioinformatics analysis, we selected proteins involved in the most significantly altered ferroptosis processes for validation. Moreover, HMOX1, TFR1 and SLC3A2 were selected as potential biomarkers for validation based on their potential biological functions and significantly differential expression, as well as their expression change ploidy, and two unique peptides were used for each protein. The PRM was performed using peak area quantification. The peak area distribution of the fragment ions of the selected peptides for the three target proteins is shown below (Figure 11). The target peptides of the target proteins were analyzed via Skyline software (v.3.6), the relative expression differences of the target proteins in different sample groups were further calculated based on the relative expression of the corresponding peptide of each target protein between different sample groups and compared to the TMT results, and the PRM validation results were consistent with the trend of the TMT relative quantitative results.

### 2.7. Western Blot Analysis

Compared to the control group, the expression levels of TFR1, HMOX1 and SLC3A2 in PC12 cells treated with OGD/R were significantly increased in the model group; compared to the model group, the expression of TFR1, HMOX1 and SLC3A2 in cells treated with 6.25 µM, 12.5 µM and 25 µM Iristectorin B decreased 24 h later, and the expression of Edaravone in the positive group also decreased significantly (Figure 12), which indicates that Iristectorin B can reduce the expression of TFR1, HMOX1 and SLC3A2 in OGD/R cells, thereby playing a role in protecting stroke by inhibiting ferroptosis.

## 3. Discussion

Proteomics provides additional support for current advances in biology and is better suited to the analysis of a wide range of proteins, producing more accurate and efficient expression results [17]. Proteins involved in disease pathology are emerging as important biomarkers for a variety of diseases in biomedical experiments [18,19]. TMT-labeled nanoscale LC-MS/MS is an emerging quantitative method [20,21,22,23] used for the study of anti-stroke proteomics [24,25]. The key differential proteins that were screened for involvement in ferroptosis processes are HMOX1, TFR1, SLC3A2 and NCOA4. These target proteins were further validated via parallel reaction monitoring (PRM) [26,27,28,29,30], the roles of HMOX1, TFR1 and SLC3A in ferroptosis were successfully validated via PRM, and the overall trend of PRM validation results and TMT experimental results were in high agreement, which supported the proteomics data rationality and reliability. Therefore, the identified differentially expressed proteins became the focus of further research into disease resistance mechanisms.

Stroke is an acute cerebrovascular disease that is regulated via multiple cell death mechanisms [31]. During a stroke, the blood supply is subsequently blocked, producing a series of pathological responses leading to different types of cell death, including apoptosis, autophagy, cell necrosis and ferroptosis [32]. Among other things, ferroptosis is an iron-dependent programmed cell death pathway that plays an important role in the pathology of stroke [33]. Ischemia/reperfusion of the brain causes severe cellular damage and death, a process that is exacerbated by ferroptosis [34].

Iristectorin B protects against acute stroke-induced neuronal death by inhibiting ferroptosis. Proteomic results showed that TFR1 upregulation indirectly caused an increase in NCOA4 expression through the regulation of the ferritin FTH1. Transferrin receptor protein 1 (TFR1) is a transmembrane glycoprotein that also acts as a cellular iron regulator responsible for iron transport at the cell membrane and plays an important role in maintaining iron homeostasis and regulating ferroptosis in the brain [35]. When cells are damaged by ischemia and hypoxia in stroke, TFR1 becomes dysfunctional, resulting in an imbalance in iron homeostasis, and the presence of excess Fe^2+^ in the cell triggers a Fenton reaction that produces large amounts of ROS and lipid peroxidation products, ultimately causing cellular ferroptosis [36]. The transferrin (TF)-TFR1 signaling pathway is the main pathway for iron uptake by neurons. After ischemia in brain tissue, the expression level of TFR1 increases, and a large amount of iron ions are transferred from extracellular to intracellular, which can lead to intracellular iron overload [37].

Extracellular free Fe^3+^ forms a complex with the transferrin TF, which binds to the transferrin receptor TFR1 on the cell membrane and is endocytosed to form endosomes for transport into the cell. Fe^3+^ is then reduced to Fe^2+^ by the action of the six-transmembrane epithelial antigen of prostate 3 (STEAP3), followed by Fe^2+^ binding to the divalent metal-ion transporter-1 (DMT1). DMT1 mediates the transport of Fe^2+^ from endosomes to cytoplasmic lysates [38,39]. On one hand, Fe^2+^ can be pumped through membrane iron transport proteins in the cell membrane, and on the other hand, Fe^2+^ can be stored in cytoplasmic ferritin to achieve intracellular iron homeostasis [40]. Ferritin, composed of FTH1 and FTL, has the ability to chelate and store iron and plays an important antioxidant role in the cell, thereby protecting it from oxidative stress [41]. When excess Fe^2+^ is produced in cells, it can directly catalyze the production of lipid ROS through the Fenton reaction, eventually causing a gradual increase in intracellular ROS levels and triggering ferroptosis [42].

Heme oxygenase-1 (HMOX1) degrades hemoglobin to produce Fe^2+^ to increase Fe^2+^ levels and subsequently increase the occurrence of ferroptosis in cells [43,44]. After the oxidation of intracellular Fe^2+^ to Fe^3+^, Fe^3+^ forms iron stores with ferritin, which subsequently binds to nuclear receptor coactivator 4 (NCOA4) and delivers ferritin conjugates to autophagosomes [45,46], where ferritin degradation releases large amounts of iron, leading to a climb in intracellular iron levels. NCOA4 is a selective receptor for ferritin autophagy and when bound to ferritin, NCOA4 can mediate ferritin autophagy to increase Fe^2+^ and cause ferroptosis [47].

The cystine-glutamate retrotransporter (System Xc-) is an antioxidant system present in the cell membrane, forming a heterodimer consisting of two subunits of the transmembrane transporter protein SLC7A11 and the regulatory protein SLC3A2 (also known as CD98) linked by disulfide bonds, mediating the exchange in intracellular glutamate with extracellular cystine to produce GSH and GPX4, which has a positive effect on the inhibition of ferroptosis [48,49,50]. The cystine transported into the cell is reduced to cysteine, which is involved in the synthesis of reduced glutathione (GSH), an essential cofactor influencing the activity of GPX4, which is limited by the limited concentration of cysteine in the cell, thus limiting the rate of GSH synthesis [51]. GPX4 is a key regulator of the ferroptosis process, converting GSH to oxidized glutathione (glutathioneoxidized, GSSG) and reducing lipid peroxides (L-OOH) to alcohols (L-OH) to reduce ROS levels and avoid damage to cell membranes caused by lipid peroxides [52]. When the extracellular glutamate content is too high, it inhibits the transport of extracellular cystine by System Xc-, leading to the disruption of intracellular cysteine metabolism, inhibiting System Xc-, resulting in GSH depletion indirectly affecting the activity of GPX4, leading to abnormal accumulation of intracellular lipid peroxides, which, in turn, induces ferroptosis [53,54,55]. It has also been shown that heatshockprotein 90 (HSP90) induces programmed necrosis and ferroptosis by promoting RIPK1 phosphorylation and inhibiting GPX4 activation [56]. 

In conclusion, HMOX1, TFR1 and SLC3A2 act as potential biomarkers of a stroke and play an important role in the underlying mechanism of stroke through the activation of ferroptosis. The present work provides new evidence for the anti-stroke effect of the natural product isoflavone Iristectorin B on ferroptosis and suggests that Iristectorin B may be a potential therapeutic agent for stroke.

## 4. Materials and Methods

### 4.1. PC12 Cell Culture

PC12 cells (Chinese Academy of Sciences, Shanghai, China) were cultured in high-sugar DMEM medium containing 10% fetal bovine serum and 1% penicillin and streptomycin in 5% CO_2_ at 37 °C for 3–4 d. PC12 cells were digested and passaged with 0.25% trypsin when they reached about 70–80% wall attachment. The cells were then inoculated into 96-well plates at 100 μL per well and incubated at 5% CO_2_ and 37 °C for 1–2 d. The cells were then used for subsequent experiments when the cell density reached 80–90%.

### 4.2. Establishment and Experimental Grouping of a Glucose Oxygen Deprivation/Reoxygenation (OGD/R) Ex Vivo Stroke Model

To investigate the protective effect of the natural isoflavone Iristectorin B on PC12 after OGD/R treatment injury, PC12 cells treated as described above were divided into the following groups: Control—no OGD/R injury treatment. Model—100 μL of sugar-free EBSS solution containing 10 mmol/L Na_2_S_2_O_4_ was added and acted for 2 h. The cells were replaced with normal medium for reoxygenation and continued to be cultured for 24 h, constituting an induced OGD/R injury model to simulate ischemic brain injury. OGD/R was supplemented with normal culture medium containing Iristectorin B 6.25 µM, 12.5 µM and 25 µM for 24 h. OGD/R was supplemented with 10 µM Edaravone as a positive control (*n* = 6). The natural isoflavone Iristectorin B, EBSS, Na_2_S_2_O_4,_ Edaravone were sourced from Sigma-Aldrich, located in St. Louis, MO, USA.

#### 4.2.1. Morphological Observation

PC12 cells were incubated for 48 h at 37 °C after the addition of treatment factors and observed under an inverted microscope for morphological changes in the above treatment groups (*n* = 6).

#### 4.2.2. MTT Method to Measure Cellular Activity

After the drug action, 10 µL of MTT (Sigma-Aldrich, located in St. Louis, MO, USA) at a concentration of 5 g/L was added to each well in a 96-well cell culture plate under light-proof conditions, and the supernatant was discarded after incubation for 3–4 h at 37 °C with 5% CO_2_. Next, 100 µL of DMSO (Sigma-Aldrich, located in St. Louis, MO, USA) was added to each well, and the absorbance of each well was measured at 490 nm using an enzyme standard for 10–15 min (*n* = 6). 

#### 4.2.3. Detection of Ca^2+^ Content in PC12 Cells

After the drug effect was finished, the culture medium loaded with a 2.5 µmol/L Fluo-3AM fluorescent probe (Sigma-Aldrich, located in St. Louis, MO, USA) was incubated with the cells for 1h at 5% CO_2_ and 37 °C. After appropriate washing, the fluorescence generated by calcium ions was observed via fluorescence microscopy and detected via enzyme standardization at 488 nm and 530 nm, respectively. Meanwhile, to exclude the effect of cell density, MTT was subsequently performed to make the cell density. The final intracellular Ca^2+^ content of PC12 cells was calculated after normalization.
$$H=\frac{A}{O}÷B\times 100\%$$

H is the Ca^2+^ content, A is the calcium ion assay value per well, O is the absorbance of the corresponding well and MTT B is the mean MTT value for each treatment group (*n* = 6).

#### 4.2.4. LDH Release Assay in PC12 Cells

At the end of drug action, intracellular release from PC12 cells was measured according to the LDH assay kit instructions, and the degree of damage to PC12 cells was evaluated based on the LDH (Sigma-Aldrich, located in St. Louis, MO, USA) release rate.
$$D=\frac{M}{C}\times 100\%$$

D is the LDH release rate (%), M is the sample supernatant LDH activity and C is the maximum cellular LDH activity (*n* = 6).

#### 4.2.5. Detection of ROS Levels in PC12 Cells

After the drug effect, the cells were incubated with serum-free culture medium loaded with 10 μmol/L DCFH-DA probe (Sigma-Aldrich, located in St. Louis, MO, USA) for 20 min at 5% CO_2_ and 37 °C. After proper washing of the serum-free culture medium, the ROS level in PC12 cells was measured with the aid of an enzyme marker and calculated in the same way as the Ca^2+^ level described above (*n* = 6).

#### 4.2.6. Apoptosis Assay

After drug administration, PC12 cells were assayed for apoptosis via flow cytometry using the Annexin V-FITC Apoptosis Assay Kit (Sigma-Aldrich, located in St. Louis, MO, USA) (where Annexin V-FITC is fluorescein isothiocyanate-labeled phospholipid-binding protein and PI is propidium iodide). The apoptotic rate was detected via flow cytometry with FACSCanto II system (Becton Dickinson, Franklin Lakes, NJ, USA), while the percentage of apoptotic cells was calculated via FlowJo software (v10.6.2) (*n* = 3).

### 4.3. Proteomics Steps

#### 4.3.1. Protein Extraction

Following the above treatment, PC12 cells were divided into three groups: Control group—PC12 cells were grown normally without treatment. Model group—0.15 mL of Earle’s Balanced Salt Solution EBSS (without calcium, magnesium, phenol red) solution containing 10 mM Na_2_S_2_O_4_ was added and reacted for 1 h. The medium was replaced with normal medium for reoxygenation and cultured for a further 24 h, constituting an induced OGD/R injury model [57] to simulate ischemic brain injury. Experimental group—OGD/R with 0.15 mL of normal medium containing 25 µM Iristectorin B was reoxygenated for 24 h. 

In each of the three groups, 4× the reaction volume of lysis buffer (8 M urea, 1% protease inhibitor cocktail, 3 µM TSA, 50 mM NAM) was added and sonicated. The remaining debris was removed via centrifugation at 12,000× *g* at 4 °C for 10 min. Finally, the supernatant was transferred to a new centrifuge tube for protein concentration determination using the BCA kit (Sigma-Aldrich, located in St. Louis, MO, USA).

#### 4.3.2. Trypsin Digestion

Equal amounts were taken from each group for enzymatic digestion [58], equal amounts of standard proteins were added and the volumes were adjusted to account for lysis solution. A final concentration of 20% TCA was added, and the mixtures were vortexed and then precipitated for 2 h at 4 °C. The solutions were then centrifuged for 5 min at 4500× *g*, and the supernatant was discarded. The precipitate was washed 2–3 times with pre-cooled acetone. After drying the precipitate, TEAB was added for a final concentration of 200 mM, the precipitate was broken up via sonication, and then trypsin was added at a ratio of 1:50 (protease:protein, m/m) and left to digest for 12 h. Dithiothreitol (DTT) was added to a final concentration of 5 mM and reduced for 30 min at 56 °C. Iodoacetamide (IAA) was then added to a final concentration of 11 mM and incubated for 15 min at 25 °C and protected from light. 

#### 4.3.3. Quantitative Protein Mass Spectrometry TMT Labeling

Trypsin digested peptides were desalted with Strata X C18 (Phenomenex, Torrance, CA, USA) and vacuum freeze-dried. The peptides were lysed with 0.5 M TEAB and labeled according to the TMT kit (Thermo Fisher Scientific, Waltham, MA, USA). The labeling reagent was thawed and dissolved in acetonitrile, mixed with the peptide and incubated at 25 °C for 2 h. The labeled peptide was mixed, desalted and freeze-dried under vacuum.

#### 4.3.4. Classification

The peptides were graded via high pH reversed-phase HPLC on an Agilent 300Extend C18 column (5 µm particle size, 4.6 mm inner diameter, 250 mm length). The peptide gradient was 8–32% acetonitrile, pH 9; 60 fractions were separated in 60 min. The peptides were then combined into six fractions, and the combined fractions were vacuum freeze-dried for subsequent experiments. 

#### 4.3.5. Liquid Chromatography-Mass Spectrometry (LC-MS/MS) Analysis

The peptides were dissolved in phase A of the liquid chromatography mobile phase and separated using an UltiMate 3000 Ultra Performance Liquid Chromatography system (Thermo Fisher Scientific, Waltham, MA, USA). Mobile phase A was an aqueous solution containing 0.1% formic acid and 2% acetonitrile; mobile phase B was an aqueous solution containing 0.1% formic acid and 90% acetonitrile. The liquid phase gradient was set as follows: 0–3 min, 3–10% B; 3–8 min, 10–25% B; 8–52 min, 25–35% B; 52–56 min, 35–80% B. The flow rate was maintained at 450 nL/min. The peptides were separated using an UHPLC system, injected into an NSI ion source for ionization and then injected into an Orbitrap Exploris^TM^ 480 mass spectrometer for analysis. The ion source voltage was set at 2.0 kV, and the FAIMS compensation voltage (CV) was set at −45 V. Both the peptide parent ions and their secondary fragments were detected and analyzed using a high-resolution Orbitrap. The primary mass spectrometry scan range was set at 400–1200 *m*/*z*, and the scan resolution was set at 60,000; the secondary mass spectrometry scan range had a fixed starting point of 110 *m*/*z*, the secondary scan resolution was set at 15,000 and TurboTMT was set to TMT Reagents. The data acquisition mode used a data-dependent scanning (DDA) procedure, whereby the top 25 peptide parent ions with the highest signal intensities were selected to be sequentially fragmented into the HCD collision cell using 35% fragmentation energy after the primary scan, before being similarly sequentially analyzed for secondary mass spectrometry. To improve the effective utilization of the mass spectra, the automatic gain control (AGC) was set at 100%, the signal threshold was set at 5 × 10^4^ ions/s, the maximum injection time was set at Auto and the dynamic exclusion time of the tandem mass spectrometry scan was set at 30 s to avoid repeated scans of the parent ions. 

#### 4.3.6. Proteomics Data Analysis

Experimental secondary mass spectrometry data were searched using Proteome Discoverer (v2.4.1.15). Search parameters were set as follows: the database was Rattus_norvegicus_10116_PR_20210721.fasta (29,934 sequences), inverse libraries were added to calculate the false positive rate (FDR) due to random matches and common contamination libraries were added to the database to remove the effect of contaminating proteins from the identification results. 

The enzymatic cleavage method was set at Trypsin (Full), the number of missed cut sites was set at 2, the minimum peptide length was set at 6 amino acid residues, the maximum number of peptide modifications was set at 3, the mass error tolerance of the primary parent ion was set at 10 ppm and the mass error tolerance of the secondary fragment ion was set at 0.02 Da. Carbamidomethyl (C), TMT6plex (peptide N-Terminus), TMT6plex (K) were set as fixed modifications, and Acetyl (protein N-Terminus) and Oxidation (M) were set as variable modifications. The quantification method was set at TMT-10 plex, and the FDRs for protein, peptide and PSM identification were all set at 1%. The accuracy FDR for identification at the spectral, peptide and protein levels was set at 1%; identified proteins needed to contain at least one specific peptide, with average ratio-fold change >1.3 (upregulation) and <0.77 (downregulation), as well as a *p*-value < 0.05.

#### 4.3.7. Bioinformatics Analysis 

The network for GeneOntology (GO) mapping and annotation was generated and analyzed using Clu GO from Cytoscape 3.7.2. DEPs pathway enrichment analysis was conducted via the Kyoto Encyclopedia of Genes and Genomes (KEGG) database and the Database for Annotation. *p*-value < 0.05 was defined as statistically significant. We searched for the protein genes in the IntAct molecular interaction database to obtain the mutual interaction data between proteins (PPI). The results were downloaded and imported into the Cytoscape package (version 3.2.1) in extensible graph markup and modeling language (XGMML) format to visualize and further analyze the functions of the differentially expressed proteins. In addition, the importance of the proteins in the PPI network was assessed by calculating the degree of interaction for each protein.

#### 4.3.8. PRM Validation

The tryptic peptides were dissolved in 0.1% formic acid (solvent A) directly loaded onto a home-made reversed-phase analytical column. The gradient comprised an increase in solvent B from 6 to 23% (0.1% formic acid in 98% acetonitrile) over 38 min, 23 to 35% in 14 min and climbing to 80% in 4 min before holding at 80% for the last 4 min, all at a constant flow rate of 700 nL/min on an EASY-nLC 1000 UPLC system.

The peptides were subjected to NSI source, followed by tandem mass spectrometry (MS/MS) in Q Exactive^TM^ Plus (Thermo Fisher Scientific, Waltham, MA, USA) coupled online to the UPLC. The electrospray voltage applied was 2.0 kV. The *m/z* scan range was 350 to 1000 for full scan, and intact peptides were detected in the Orbitrap at a resolution of 35,000. Peptides were then selected for MS/MS using NCE setting at 27, and the fragments were detected in the Orbitrap at a resolution of 17,500. The data-independent procedure alternated between one MS scan and 20 MS/MS scans. Automatic gain control (AGC) was set at 3 × 10^6^ for full MS and 1 × 10^5^ for MS/MS. The maximum IT was set at 20 ms for full MS and auto for MS/MS. The isolation window for MS/MS was set at 2.0 *m*/*z*.

The resulting MS data were processed using the software Skyline (v.3.6). Peptide settings: the enzyme was set as Trypsin [KR/P], and the maximum missed cleavage was set as 2. The peptide length was set as 8–25, variable modification was set as Carbamidomethyl on Cys and oxidation on Met, and maximum variable modifications were set as 3. Transition settings: precursor charges were set as 2 or 3, ion charges were set as 1 or 2 and ion types were set as b, y or p. The product ions ranged from ion 3 to the last ion, and the ion match tolerance was set as 0.02 Da.

#### 4.3.9. Western Blot Analysis 

The preparation of protein samples, lysis of cells on ice for about 20 min, centrifugation at 4 °C and 13,000× *g* for 5 min and detection of protein concentration using BCA protein detection kit were carried out. Protein was separated using 5–15% sodium dodecyl sulfate polyacrylamide gel electrophoresis (SDS-PAGE) and transferred to a polyvinylidene fluoride (PVDF) membrane. We transferred the film at a current of 250 mA for approximately 1.5 h, sealed the membrane in 5% skimmed milk powder and shook it at low speed on a horizontal shaker at 25 °C for 1 h, and was incubated overnight at 4 °C with primary antibodies. which was subsequently followed by an additional incubation phase involving horseradish peroxidase-linked secondary antibody. The concluding steps involved the identification and quantification of protein bands present on the membranes, which was accomplished employing ECL Western blotting detection reagents and quantified utilizing ImageJ software (1.53t), sourced from the National Institutes of Health, located in Bethesda, MD, USA.

### 4.4. Data Analysis 

The data were statistically analyzed using GraphPad Prism 9 and SPSS v20.0. The significance of differences was determined via Student’s t test, and the significance level was *p* < 0.05. All data shown are the means ± standard deviations (SDs) (*n* = 3).

## 5. Conclusions

In conclusion, the present study indicated that Iristectorin B significantly reduced the damage to PC12 cells caused by OGD/R, decreased apoptosis, enhanced cell survival and reduced Ca^2+^, LDH and ROS levels. The results showed that Iristectorin B had a significant protective effect on Na_2_S_2_O_4_-injured PC12 cells, and the mechanism may be related to the protective effect of neurons in the brain. The differential proteins HMOX1, TFR1 and SLC3A2, which are associated with stroke by regulating the iron death pathway after the action of Iristectorin B, were identified using TMT relative quantification combined with PRM targeted validation technology, and the overall trend of the PRM and Western blotting validation results and the results of the TMT experiments were of high consistency. Iristectorin B may act as a protective agent against stroke by regulating ferroptosis, and SLC3A2, TFR1 and HMOX1 may serve as potential diagnostic biomarkers for stroke, providing additional evidence to support the importance of ferroptosis in stroke. Consequently, additional in vivo and clinical studies experiments are needed to verify the mechanisms disclosed by our study. Nonetheless, our study provides new strategies for further research into the anti-stroke mechanisms of the natural drug isoflavones and the development of new drugs.

## Figures and Tables

**Figure 1 ijms-24-15195-f001:**
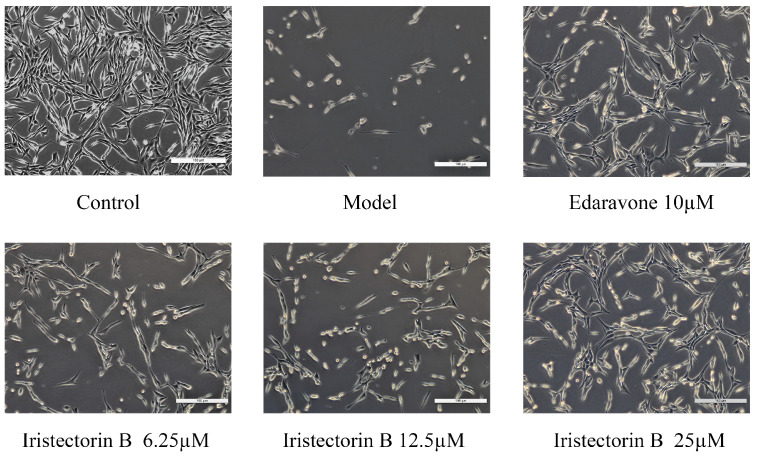
Morphological changes in PC12 cells in each group.

**Figure 2 ijms-24-15195-f002:**
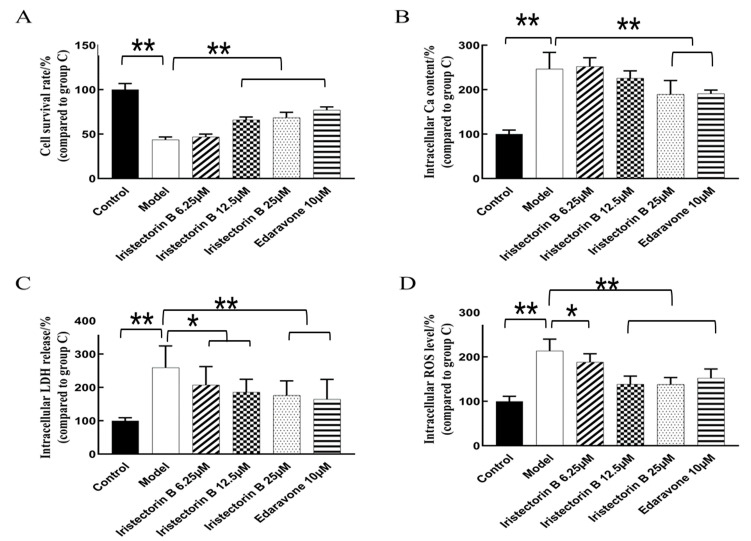
Effect of the treatment of the PC12 cells with increasing concentrations of Iristectorin B. Panel (**A**): survival rate; Panel (**B**): intracellular Ca^2+^ content; Panel (**C**): release of LDH; Panel (**D**): ROS levels (* for *p* < 0.05 and ** for *p* < 0.01) (*n* = 6).

**Figure 3 ijms-24-15195-f003:**
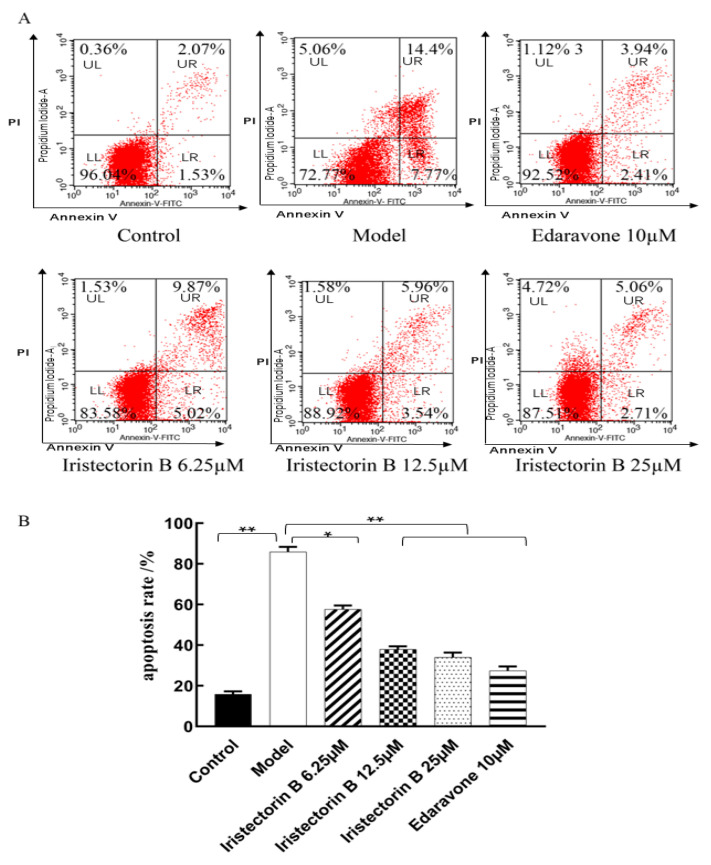
Effect of Iristectorin B on apoptosis of PC12 cells. Panel (**A**): sorting of the cells after the use of Annexin V-FITC Apoptosis Assay Kit; Panel (**B**): effect of Iristectorin B on the apoptosis rate (* for *p* < 0.05 and ** for *p* < 0.01) (*n* = 3).

**Figure 4 ijms-24-15195-f004:**
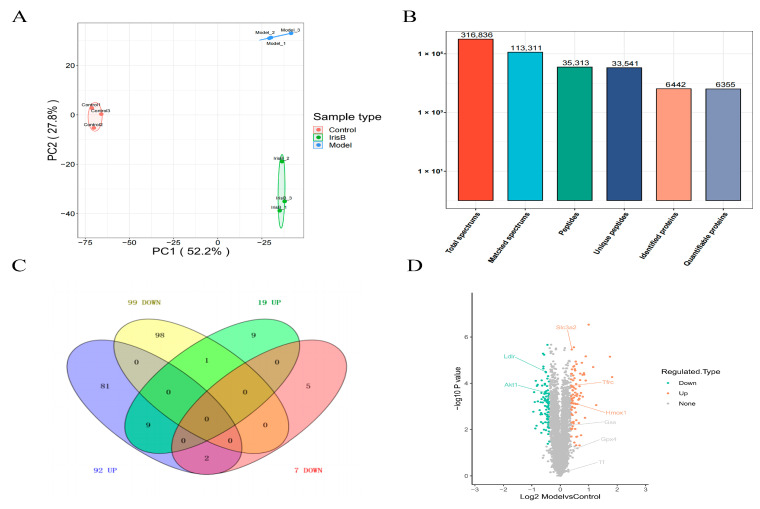
Statistical map of differentially expressed proteins. Panel (**A**): principal component analysis graph. Panel (**B**): statistical graph of mass spectrometry. Panel (**C**): Venn diagram. Panel (**D**): volcano plot.

**Figure 5 ijms-24-15195-f005:**
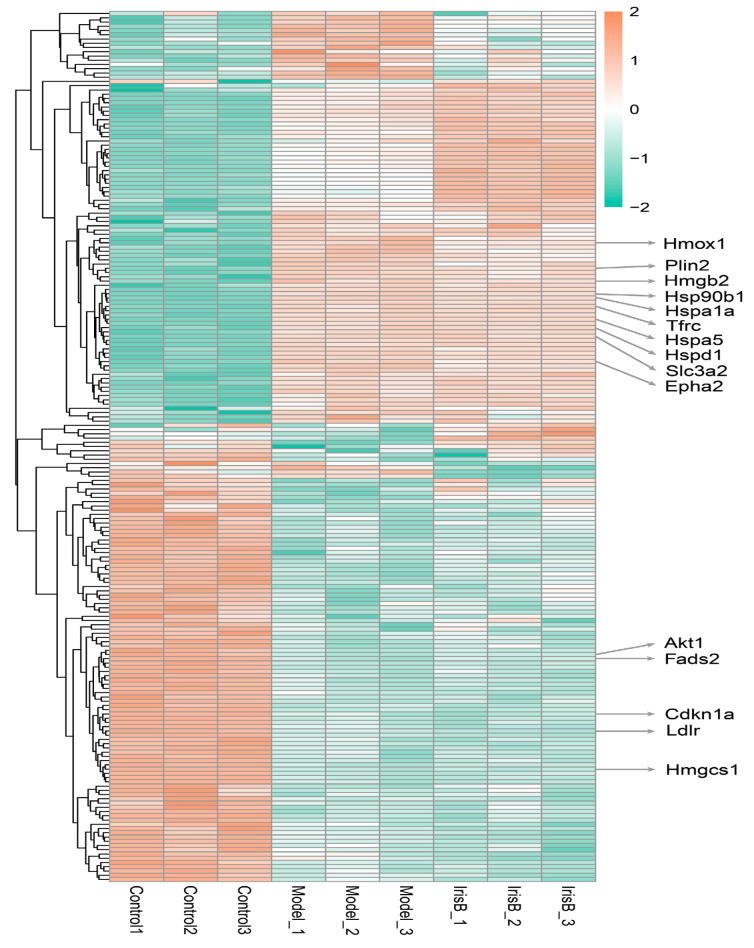
Heat map.

**Figure 6 ijms-24-15195-f006:**
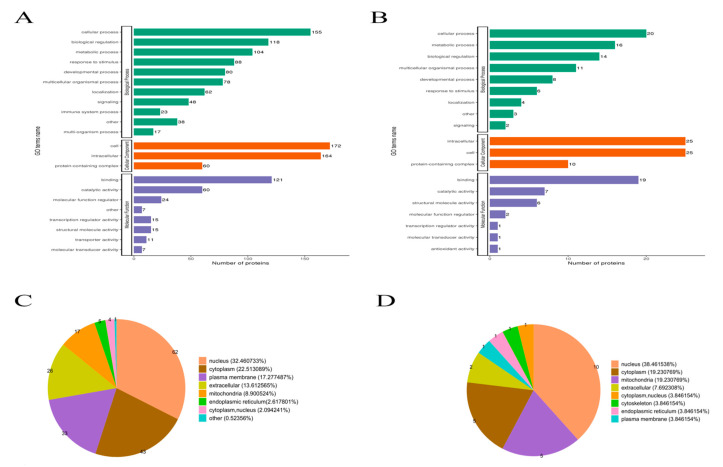
GO secondary annotation classification chart. Panel (**A**): model group/control group. Panel (**B**): experimental group/model group. Panel (**C**): classification diagram of subcellular structure localization model/control group. Panel (**D**): experimental group/model group.

**Figure 7 ijms-24-15195-f007:**
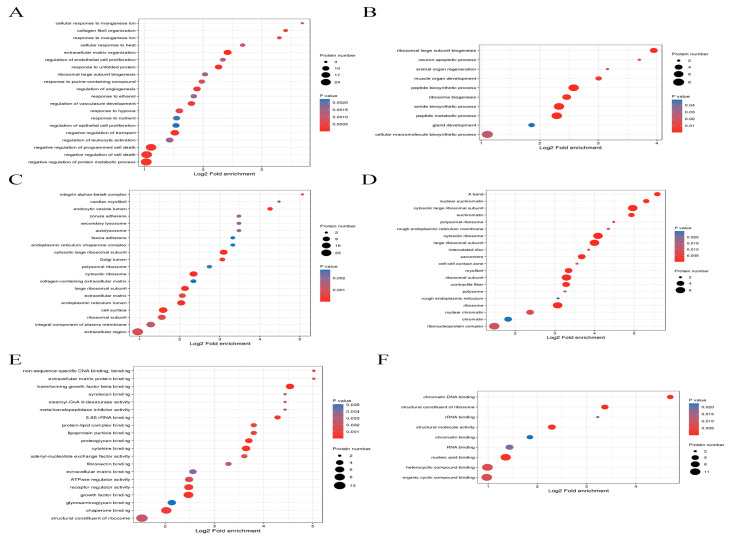
Bubble diagram for differential protein enrichment analysis of biological processes. Panel (**A**): model group/control group. Panel (**B**): experimental group/model group. Panel (**C**): bubble diagram for differential protein enrichment analysis of cellular fractions model group/control group. Panel (**D**): experimental group/model group. Panel (**E**): bubble diagram for molecular functional difference protein enrichment analysis model group/control group. Panel (**F**): experimental group/model group.

**Figure 8 ijms-24-15195-f008:**
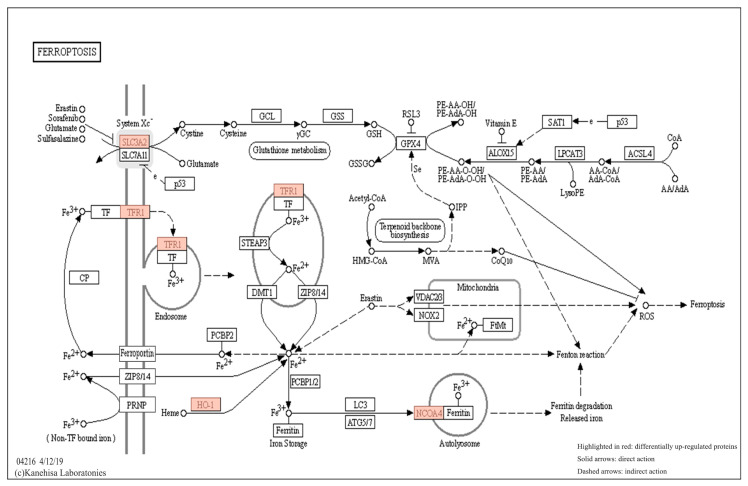
KEGG Ferroptosis pathway diagram.

**Figure 9 ijms-24-15195-f009:**
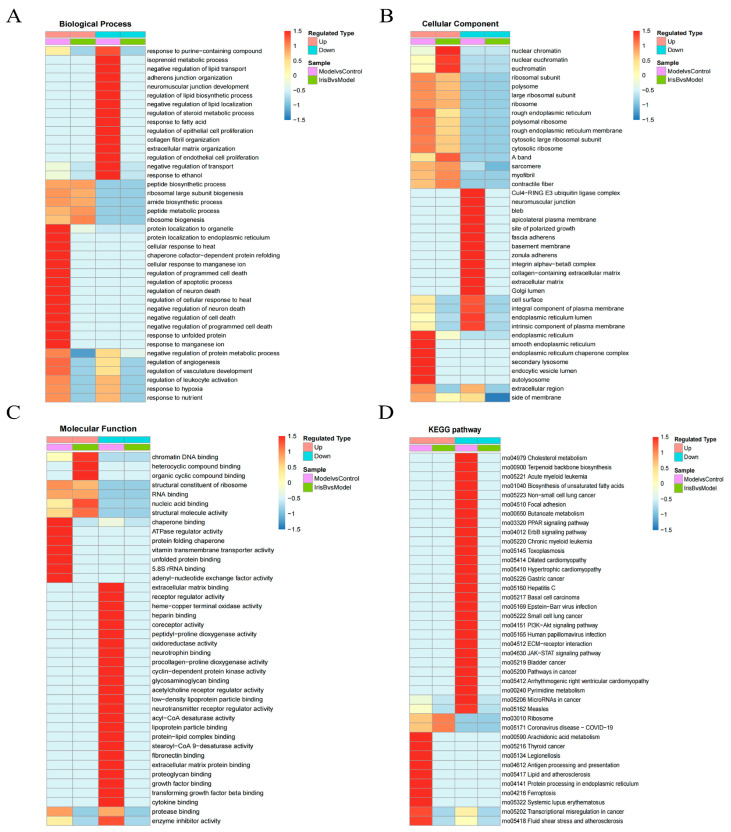
Heat map. Panel (**A**): biological. Panel (**B**): cellular component. Panel (**C**): molecular function. Panel (**D**): heat map of KEGG.

**Figure 10 ijms-24-15195-f010:**
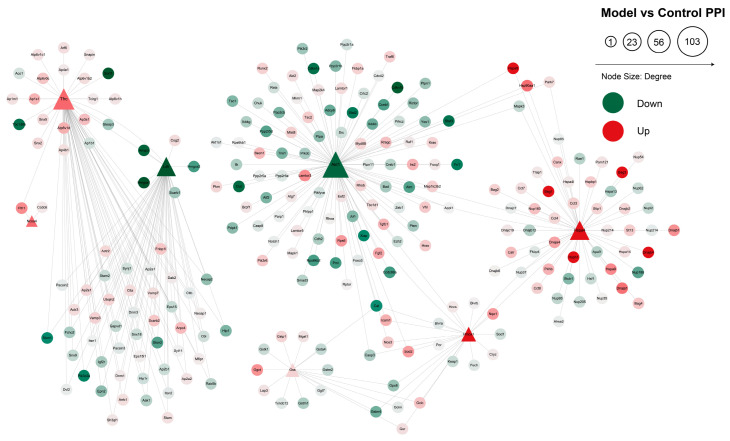
Protein interaction network.

**Figure 11 ijms-24-15195-f011:**
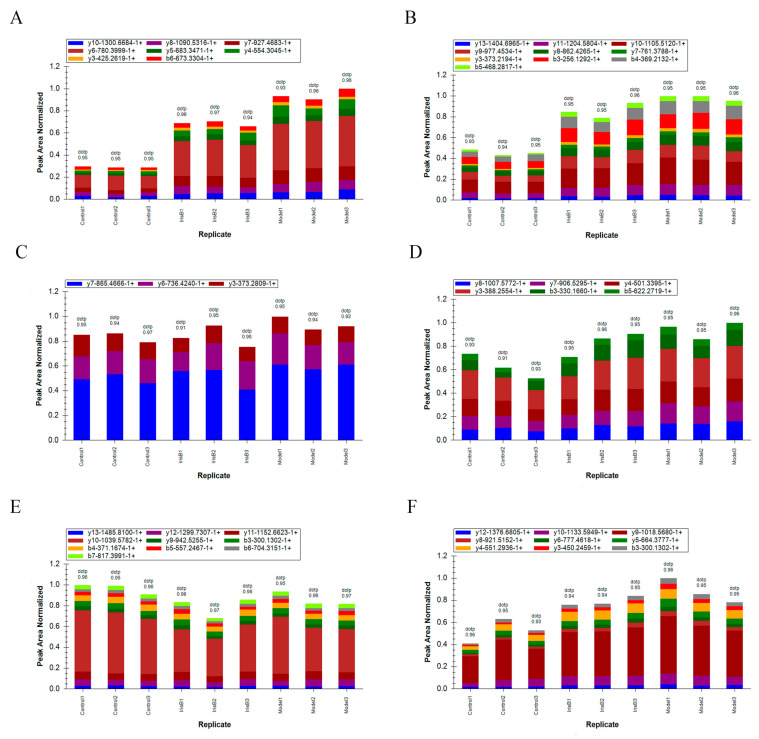
Ion peak area distribution. Panel (**A**): peak area distribution of fragment ions for peptide QNPVYAPLYFPEELHR (corresponding to protein P06762-HMOX1). Panel (**B**): peak area distribution of fragment ions for peptide PASLVQDTTSAETPR (corresponding to protein P06762-HMOX1). Panel (**C**): peak area distribution of fragment ions for peptide LNSIEFTDIIK (corresponding to protein Q99376-TFR1). Panel (**D**): peak area distribution of fragment ions for peptide LDTYEILIQK (corresponding to protein Q99376-TFR1). Panel (**E**): peak area distribution of the fragment ion of peptide GQNAWFLPPQADIVATK (corresponding to protein Q794F9-SLC3A2). Panel (**F**): peak area distribution of fragment ions for peptide GQNEDPGSLLTQFR (corresponding to protein Q794F9-SLC3A2).

**Figure 12 ijms-24-15195-f012:**
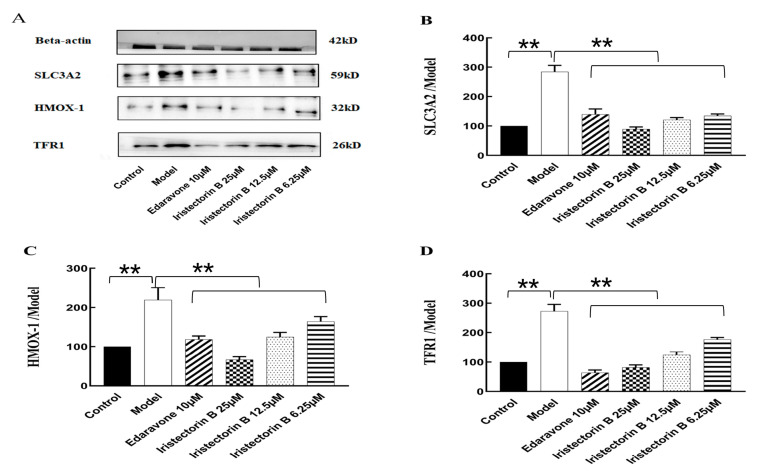
Iristectorin B effect on OGD/R protein expression. Panel (**A**): Western blotting plot. Panel (**B**): SLC3A2 protein expression compared with model. Panel (**C**): HMOX-1 protein expression with model. Panel (**D**): TFR1 protein expression with Model (** for *p* < 0.01) (*n* = 5).

**Table 1 ijms-24-15195-t001:** COG/KOG functional classification statistics for the number of differentially expressed proteins.

Group	Post-Translational Modifications	Signaling Function Proteins	Metabolism-Related Proteins	Information Storage	Unknown Function Proteins
model/control	26	71	31	36	24
experimental/control	3	8	2	12	2

**Table 2 ijms-24-15195-t002:** Differential protein enrichment analysis of biological processes.

BP	Model/Control *p* < 0.0005	BP	Experimental/Control *p* < 0.01
negatively regulated for programmed cell death	17–24	peptide biosynthesis process	6–8
negatively regulated related proteins for protein metabolic processes	17–24	amide biosynthesis process	6–8
related proteins	10–17	peptide metabolism process	6–8
cell death	17–24	ribosome synthesis ranged	4–6

**Table 3 ijms-24-15195-t003:** Differential protein enrichment analysis of molecular functions.

CC	Model/Control *p* < 0.0001	CC	Experimental/Control *p* < 0.05
cellular subunit-associated	9–16	ribosome-associated	4–6
endoplasmic reticulum lumen-associated	<9	ribosomal subunit-associated	6–8
large ribosomal subunit-associated	<9	ribosomal protein complex-associated	6–8
extracellular region-associated	16–23		

**Table 4 ijms-24-15195-t004:** Differential protein enrichment analysis of cellular components.

MF	Model/Control *p* < 0.0001	MF	Experimental/Control *p* < 0.05
transforming growth factor-β-binding proteins	4–6	nucleic acid-binding proteins	8–11
growth factor-binding proteins	6–8	structural molecular activity	5–8
cytokine-binding proteins	<4	chromatin DNA-binding proteins	2–5
ATP activity-regulating proteins	4–6	organic cyclic compound-binding proteins	8–11

## Data Availability

The data that support the findings of this study are available within the article.

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
