# Peer review of "TMT and PRM Based Quantitative Proteomics to Explore the Protective Role and Mechanism of Iristectorin B in Stroke"

_ijms, 2023, doi:10.3390/ijms242015195_

Round 1
Reviewer 1 Report
Please see the attached file

Author Response
Dear Editors and Reviewers:
Thank you for your letter and concerning our manuscript entitled“TMT and PRM based quantitative proteomics toexplore the protective role and mechanism oflristectorin B in stroke” Those comments are allvaluable and very helpful for revising andimproving our paper, as well as the importantguiding significance to our researches. We havestudied comments carefully and have madecorrection which we hope meet with approval.Revised portion are marked in red in the paper. Themain corrections in the paper and the responds tothe reviewer's comments are as flowing:
- Response tocomment: Corrections have been made as suggested by the reviewer.
- Response tocomment: Corrections have been made as suggested by the reviewer.
- Response tocomment: Corrections have been made as suggested by the reviewer.
- Response tocomment: Corrections have been made as suggested by the reviewer.
After drug administration, using the Annexin V-FITC Apoptosis Assay Kit (where Annexin V-FITC is fluores-cein isothiocyanate-labelled phospholipid-binding protein and PI is propidium iodide).The apoptotic rate was detected by flow cytometry with FACSCanto II system (Becton Dickinson), while the percentage of apoptotic cells was calculated with FlowJo software. The experiment was repeated three times. The data were statistically analyzed using GraphPad Prism 9 and SPSS v20.0. The significance of differences was determined by Student’s t test, and the significance level was p < 0.05. All data shown are the means±standard deviations (SDs) (n = 3).
Figure 3 B is a bar graph made after three experiments were detected using flow cytometry and analysed according to their data processing, and Figure 3 A shows the results of one of them.
- Response tocomment: Corrections have been made as suggested by the reviewer.
- Response tocomment: Corrections have been made as suggested by the reviewer.
- Response tocomment: Corrections have been made as suggested by the reviewer.
In addition, some of the figures (Figures 3 and 12) and figure notes (Figures 3,4,6,7,9 and 11) in the manuscript have been similarly modified in the style of Figure 2 and its figure notes.Revised portion are marked in red in the paper.
Special thanks to you for your good comments

Reviewer 2 Report
The research study titled "TMT and PRM-based Quantitative Proteomics to Investigate the Protective Effects and Underlying Mechanisms of Iristectorin B in Stroke" focuses on creating an in vitro stroke injury model using PC cells, specifically through oxygen-glucose deprivation (OGD). This model serves to assess the effectiveness of Iristectorin B, sourced from Dried Shoot, in treating stroke-related injuries in ejecta stem. The findings reveal that Iristectorin B substantially mitigates the harm inflicted on PC12 cells by oxygen glucose deprivation/reoxygenation (OGD/R). It also lowers apoptosis rates, boosts cell survival, and decreases Ca2+, LDH, and ROS levels. The study suggests that Iristectorin B significantly protects PC12 cells from damage induced by Na2S2O4, with the underlying mechanism potentially linked to the neuroprotective effects on brain neurons.
Advanced methodologies were seamlessly integrated for protein extraction and multiple analyses to investigate the quantitative proteomes across the different groups. The paper is well-crafted and offers intriguing insights. What do the authors envision as future research directions? The number of samples should be higher for strength the results.
Author Response
Thank you for your valuable comments! In terms of future research perspectives, we would like to conduct an in-depth study on the screened targets to investigate the mechanism of Iristectorin B's protective effect on stroke-induced neurological damage at the animal level, meanwhile, combined proteomics and metabolomics at the clinical level to investigate and confirmed that HMOX1, TFR1 and SLC3A2 as biomarkers of stroke! Subsequent studies with larger sample sizes will be conducted to enhance persuasiveness
Round 2
Reviewer 1 Report
The reviewer would like to thank the authors who positively answered to almost all their suggestions, improved the presentation of the results and completed the conclusions.
The reviewer would like to understand why the order of the authors was changed and why for the last author the sign “&” was used!
Some figures are yet not well dimensioned and seems stretched downward or horizontally. Please improve the final layout.
Author Response
Dear reviewer:
Thank you for your concerning our manuscript entitled “Tandem Mass Tag and Parallel Reaction Monitoring based quantitative proteomics to explore the protective role and mechanism of Iristectorin B in stroke”(IJMS-2606736). We have found that your comments are very helpful, and changes have been made accordingly. The words in red with strikethrough are the revised in the manuscript. The main corrections in the paper and the responds to the reviewer's comments are as flowing:
1.Response to comment: The present manuscript was jointly completed by Professor Zheng Meizhu and students such as Zhou Mi and Lu Tingting. After Zhou Mi graduated with a master's degree, all of the work in improving data and processing images were done by Lu Tingting in the later stage. Therefore, considering that Lu Tingting has done more than 75% of the work in this manuscript later stage, and this article is very important for the student's graduation, academic studies, and the further pursuit of a doctoral degree. We decided to adjust the author's order after the communication with all co-authers. At the same time, we also asked the editor for his advice about the adjustment of the author order. The editor suggested that we still maintain the original author order. However, due to the response time given by the editor falling behind the time we revised and uploaded the manuscript, the author order was not adjusted in time for the first revision. Therefore, in the second revision of this manuscript, we will follow the suggestions from the editor and you, and the author order will not be adjusted and kept as the initial version. We are so sorry for that our decision brought the inconvenience to you during the article revision process. Please let us know if you have any comments and suggestions about this manuscript.
2.Response to comment: We have adjusted the clarity and size of the images in the article, and uploaded clearer images, which are still clear when enlarged, and all of which have a dpi of 300 or higher.
Special thanks to you for your valuable comments!
Reviewer 2 Report
The authors have performed the suggestions.
Author Response
Special thanks to you for your valuable comments!